# DVF: Advancing Robust and Accurate Fine-Grained Image Retrieval with Retrieval Guidelines

## ABSTRACT

Fine-grained image retrieval (FGIR) is to learn visual representations that distinguish visually similar objects while maintaining generalization. Existing methods propose to generate discriminative features, but rarely consider the particularity of the FGIR task itself. This paper presents a meticulous analysis leading to the proposal of practical guidelines to identify subcategory-specific discrepancies and generate discriminative features to design effective FGIR models. These guidelines include emphasizing the object (**G1**), highlighting subcategory-specific discrepancies (**G2**), and employing effective training strategy (**G3**). Following **G1** and **G2**, we design a novel Dual Visual Filtering mechanism for the plain visual transformer, denoted as DVF, to capture subcategory-specific discrepancies. Specifically, the dual visual filtering mechanism comprises an object-oriented module and a semantic-oriented module. These components serve to magnify objects and identify discriminative regions, respectively. Following **G3**, we implement a discriminative model training strategy to improve the discriminability and generalization ability of DVF. Extensive analysis and ablation studies confirm the efficacy of our proposed guidelines. Without bells and whistles, the proposed DVF achieves state-of-the-art performance on three widely-used fine-grained datasets in closed-set and open-set settings.

## CCS CONCEPTS

• **Do Not Use This Code → Generate the Correct Terms for Your Paper**; *Generate the Correct Terms for Your Paper*; Generate the Correct Terms for Your Paper; Generate the Correct Terms for Your Paper.

## KEYWORDS

Fine-grained image retrieval,visual filtering,retrieval guidelines

## 1 INTRODUCTION

Fine-grained image retrieval (FGIR) aims to retrieve images with the same subcategory as the query images from a database within the metacategory (*e.g.*, birds, and cars) [4, 17, 39, 41, 42]. Retrieving visually similar images, however, faces challenges arising from subtle inter-class differences caused by similar objects, as well as significant intra-class variations due to factors such as location, scale, and deformation. Additionally, existing works [1, 13, 16, 24, 27, 40]

**Unpublished working draft. Not for distribution.**

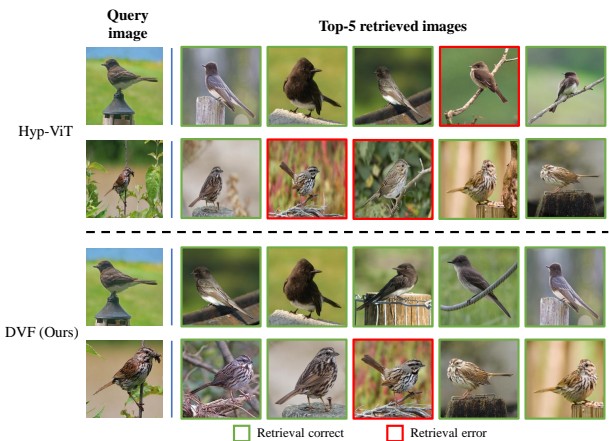

Figure 1: **We compare our method, which adheres to the guidelines, with the state-of-the-art Hyp-ViT, which violates some of these guidelines. Hyp-ViT makes retrieval errors in cases of similar images due to the objects being too small to extract subcategory-specific discrepancies. In contrast, our DVF succeeds by enlarging the objects.**

follow a closed-set training setting, where all the subcategories in the training set are known. However, the evaluation includes a closed-set setting with known test set categories and an open-set setting with unknown test set subcategories. Consequently, the essence of FGIR tasks lies in learning discriminative and generalizable embeddings to identify visually similar objects.

Recently, FGIR methods can be categorized into two main groups: encoding-based [6, 18, 24, 31] and localization-based [22, 35–37, 39, 44]. The encoding-based methods primarily optimize image-level features, which often include background and non-discriminative information. Localization-based methods typically formulate an effective object feature extraction strategy in the image encoder to capture subtle differences among subcategories, often rooted in the distinctive properties of object parts. However, these methods still suffer from small-sized objects in the input image, making it difficult to identify discriminative regions. Moreover, the scarcity of fine-grained image data jeopardizes the model's discriminative capacity and generalization ability.

Based on the preceding analysis, we present a set of guidelines for designing high-performance fine-grained image retrieval models:

- **G1: emphasize the object.** The FGIR models should utilize object-emphasized images as input to alleviate the challenges posed by small-sized objects that make identifying objects and discerning subcategory-specific discrepancies difficult [35]. Without object-emphasized images as input, the accuracy of previous methods [35, 39] is fundamentally limited.

- **G2: highlight subcategory-specific discrepancies.** Given the considerable intra-class differences and subtle inter-class variations inherent in the FGIR task, highlighting subcategory-specific discrepancies becomes paramount. An illustration of this is seen in methods [22, 37] that do not highlight subcategory-specific discrepancies, thus restricting their discriminative capability.
- **G3: employ effective training strategy.** As shown in previous studies [46], limited fine-grained image data inevitably limits retrieval performance. Some works [12, 35] affirm that effective training strategies can alleviate this limitation. However, others still neglect this aspect, hindering the design of high-performance retrieval models.

Following the guidelines above, we propose a straightforward yet effective approach: the plain visual transformer equipped with a **D**ual **V**isual **F**iltering mechanism, referred to as DVF. Fig. 2 illustrates the proposed DVF framework, consisting of an Object-oriented Visual Filtering module (OVF) and a Semantic-oriented Visual Filtering module (SVF), to generate discriminative features. Motivated by the visual foundation model's ability to accurately detect arbitrary objects without requiring additional fine-tuning, OVF utilizes a visual foundation model and a post-processing strategy to zoom in on the object in the input image (**G1**). In this way, OVF can adjust image content to aid subcategory prediction and locate regions of subcategory-specific discrepancies. In SVF, we propose a token importance generator to calculate token-level importance, which is subsequently used as a weighting factor in class attention to enhance the selection of discriminative tokens and eliminate noisy features (**G2**). This improves the model's ability to discern subtle discrepancies between subcategories. Following **G3**, we propose a Discriminative Model Training strategy that combines data augmentation with loss constraints, significantly improving the discriminability of DVF across both closed-set and open-set scenarios.

In summary, the primary contributions of this work are as follows:

- By considering the particularity of fine-grained image retrieval, we formulate a set of practical guidelines for designing high-performance fine-grained image retrieval models.
- Following our proposed guidelines, we develop a straightforward yet potent retrieval model called DVF, which incorporates a dual visual filtering mechanism to capture subcategory-specific discrepancies. Additionally, we employ a discriminative training strategy to enhance the model's discriminability and generalization ability.
- The experimental results on three fine-grained image retrieval benchmarks demonstrate the superior performance of DVF exhibits in closed-set and open-set scenarios. Moreover, visualization results demonstrate DVF's capability to capture discriminative image regions accurately.

## 2 RELATED WORKS

Existing methods for fine-grained image retrieval (FGIR) can be categorized as encoding-based or localization-based. The encoding-based methods [6, 18, 24, 31] aim to learn an embedding space in which samples of a similar subcategory are attracted and samples

of different subcategories are repelled. The methods can be decomposed into roughly two components: the image encoder maps images into an embedding space, and the metric method ensures that samples from the same subcategories are grouped closely, while samples from different subcategories are separated. While these studies have achieved significant achievements, they primarily concentrate on optimizing image-level features that include numerous noisy and non-discriminatory details. Therefore, the localization-based methods [9, 22, 35–37, 39] are proposed, which focus on training a subnetwork for locating discriminative regions or devising an effective strategy for extracting attractive object features to facilitate the retrieval task. Unlike these approaches, our methodology considers the specific characteristics of the FGIR tasks, offering guidance for designing high-performance retrieval models.

## 3 METHOD

### 3.1 Image Encoder

For the input image $\mathbf{I} \in \mathbb{R}^{3 \times H \times W}$, the image encoder will first split it into $N = N_h \times N_w$ non-overlapping patches of size $P \times P$. Here, $N_h = \frac{H}{P}$ and $N_w = \frac{W}{P}$. Subsequently, these patches are transformed into embedding tokens $\mathbf{E} = [\mathbf{E}^1, \mathbf{E}^2, \ldots, \mathbf{E}^N] \in \mathbb{R}^{N \times D}$ using learnable linear projection, where $D$ denotes the dimension of each token. Finally, the embedding tokens $\mathbf{E}$ are concatenated with a class token $\mathbf{E}^{class} \in \mathbb{R}^D$, and the position information of the tokens is retained through combined a learnable position embedding $\mathbf{E}_{pos}$ to form the initial input token sequence as $\mathbf{E}_0 = [\mathbf{E}^{class}, \mathbf{E}^1, \mathbf{E}^2, \ldots, \mathbf{E}^N] + \mathbf{E}_{pos}$. The image encoder comprises $L$ layers, each incorporating multi-head self-attention (MHSA) and a multi-layer perception (MLP) block. Therefore, considering the input $\mathbf{E}_{i-1}$ of the $i$-th layer, the output can be expressed as follows:

$$\mathbf{E}'_i = \text{MHSA}(\text{LN}(\mathbf{E}_{i-1})) + \mathbf{E}_{i-1}, \tag{1}$$

$$\mathbf{E}_i = \text{MLP}(\text{LN}(\mathbf{E}'_i)) + \mathbf{E}'_i, \tag{2}$$

where $i = \{1, 2, \ldots, L\}$, and $\text{LN}(\cdot)$ denotes the layer normalization operation. The class token $\mathbf{E}_L^{class}$ from the $L$-th layer serves as the retrieval embedding, we simplify it to $\mathbf{E}^R$.

### 3.2 Dual Visual Filtering Mechanism

Subtle yet discriminative discrepancies are widely acknowledged as crucial for FGIR [11, 23, 29, 38]. However, in real-life scenarios, subcategory objects often occupy only a fraction of the image, with the background dominating a larger portion. This presents challenges for the model in both object recognition and identifying discriminative regions within objects. Additionally, traditional visual Transformers (ViTs) [5, 32, 33] are originally designed to represent general visual concepts, rather than effectively capturing subtle discrepancies for subordinate categories within a given category in fine-grained visual concepts. To address these challenges, following **G1** and **G2**, we design a dual visual filtering mechanism consisting of two components: an object-oriented visual filtering module and a semantic-oriented visual filtering module.

*3.2.1 Object-oriented Visual Filtering Module.* Following **G1**, we propose the Object-oriented Visual Filtering module (OVF) to zoom in on the object in the image. Previous methods [19, 30, 35, 39] typically train a sub-network to locate objects in a weakly supervised

**Figure 2: Overview of the proposed framework. The framework consists of two core modules, 1) Object-oriented Visual Filtering Module: utilizes a visual foundation model to zoom in object in the input image (details are in Section 3.2.1); 2) Semantic-oriented Visual Filtering Module: accounts for the attention of the class token as well as the importance of the embedding token itself to locate discriminative regions in the object (details are in Section 3.2.2).**

manner, consuming training resources and often leading to inaccurate positioning due to the limitations of weakly supervised training. In contrast, DVF investigates the utilization of visual foundation models [15, 21, 43] for accurate object localization in a **training-free manner**. Specifically, OVF leverages the visual foundation model Grounding-DINO [21] along with a post-processing strategy, to locate and magnify objects. This allows the model to capture more discriminative details, thereby enhancing the quality of the representation. Besides, the proposed OVF is model-agnostic, making it applicable to other FGIR methods.

Specifically, Grounding-DINO utilizes an image and its associated text prompt (metacategory name) as input, as shown in Fig. 2, to generate coordinate results and confidence scores for the corresponding metacategory object. We can use the coordinate results to crop the input image to obtain an image with emphasized objects. However, we observed that to acquire input images with emphasized objects, it is necessary to 1) confirm the presence of the object in the image, 2) ensure the object remains intact, and 3) prevent deformation after image resizing. To achieve this, we introduce a

**Algorithm 1** The process of Post-Processing in OVF
***
**Input:** Input image $X$, text prompt $P$, visual foundation model $Y$
**Output:** *result*
1:    $score, result \leftarrow Y(X, P)$
2:    **if** $score < \alpha$ **then**
3:       $reulst \leftarrow X$
4:    **else**
5:       $reulst \leftarrow FilteringAdaptation(reulst)$
6:    **end if**
***

**post-processing** strategy, which involves self-checking and filtering adaptation, the process of **post-processing** is summarized in Algorithm 1.

To confirm the presence of the object in the image, we perform self-checking before utilizing the detection results. If the detection score surpasses the threshold $\alpha$, we adopt the detection result; otherwise, the original input image serves as the output of the OVF. In this paper, we set $\alpha$ to 0.5. To ensure the object remains intact, we enlarge the detection results by a factor of 1.1, ensuring the inclusion of edge areas possibly overlooked during the detection

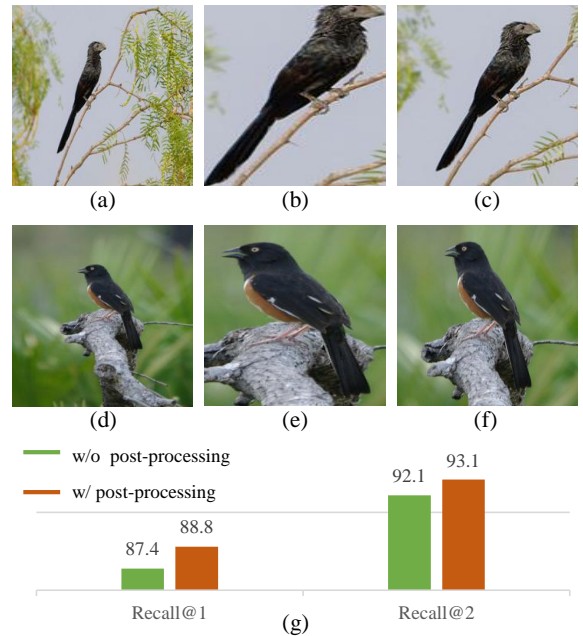

Figure 3: (a),(d) origin input image, (b),(e) object-oriented visual filtering without post-processing, (c),(f) object-oriented visual filtering with post-processing. (g) the evaluation results (%) on CUB-200-2011 with/without post-processing.

process. In addition, to prevent deformation after image resizing, we constrain the aspect ratio of the image to 3:4 by adding margins to the shortest side. The experimental results depicted in Fig. 3(g) validate the effectiveness of the post-processing strategy.

*3.2.2 Semantic-oriented Visual Filtering Module.* In plain visual transformers [5], the transformer layer treats all image regions divided into patches equally, leading to the merging of a substantial amount of redundant patch information. This limits the retrieval accuracy of ViT-based models in fine-grained image retrieval tasks. Following **G2**, we propose a Semantic-oriented Visual Filtering module (SVF). This module merges attention scores from the class token with token importance generated by a token importance generator to select discriminative tokens and eliminate noisy tokens.

The attention score of the class token indicates the semantic significance of each region for subcategory prediction. Therefore, for the attention scores $\mathbf{A} = [\mathbf{A}^1, \mathbf{A}^2, \ldots, \mathbf{A}^m, \ldots, \mathbf{A}^M]$ from the class token in the penultimate layer, where $\mathbf{A}^m \in \mathbb{R}^N$ denotes the attention score of the $m$-th attention head. We aggregate the attention scores of all heads to obtain the semantic scores for embedding tokens as $\hat{\mathbf{A}} = \sum_{i=1}^M \mathbf{A}^i$. Then, the semantic score $\hat{\mathbf{A}}$ can be used as the evaluation metric to select discriminative tokens as in previous work [11]. However, there arises the issue of inaccurate subcategory prediction, leading to the semantic score's inability to precisely reflect the semantic importance of the token. This problem is particularly pronounced in the open-set scenario, where the subcategory in the test set remains unknown. To eliminate the issue, the SVF introduces a token importance generator $\Omega$ to transform the input embedding tokens into a new space, thereby specifying the importance of the embedding tokens. Concretely, we project the embedding tokens into the token importance $\mathbf{Z} \in \mathbb{R}^N$. This can

Table 1: Ablation of our data-augmentation strategy on CUB-200-2011 in the open-set setting.

| Data-Augmentation | | | CUB-200-2011 | |
|---|---|---|---|---|
| ColorJitter | Grayscale | Gaussian Blur | Recall@1 | Recall@2 |
| ✓ | ✗ | ✗ | 88.4 | 92.8 |
| ✓ | ✓ | ✗ | 88.6 | 92.9 |
| ✓ | ✓ | ✓ | **88.8** | **93.1** |

be expressed as follows:

$$\mathbf{Z} = \sigma(\Omega(\mathbf{E}_i)) \ i = \{1, 2, \ldots, N\}, \quad (3)$$

where $\sigma(\cdot)$ is a sigmoid activation function, $\Omega$ is a linear projection layer, and $\mathbf{E}_i$ is the embedding token.

We choose the semantic score $\hat{\mathbf{A}}$ of the token as the basic evaluation metric. Through token importance weighting as per Eq. 4, we can more accurately estimate the semantic score $\mathbf{O}$.

$$\mathbf{O} = \hat{\mathbf{A}} \oplus (\hat{\mathbf{A}} \otimes \mathbf{Z}), \quad (4)$$

where $\oplus$ and $\otimes$ are element-wise summation and element-wise multiplication, respectively. Subsequently, we select the top-$k$ significant embedding tokens with high activation values, denoted as $\mathbf{ids} = \mathsf{TopK}(\mathbf{O}) \in \mathbb{Z}^K$, where $\mathsf{TopK}(\cdot)$ sorts the importance scores $\mathbf{O}$ in descending order and returns the index $\mathbf{ids}$ of the top $k$ embedding tokens. The selected $k$ embedding tokens concatenate them with the class token to form the input sequence of the $L$-th transformer layer, expressed as:

$$\mathbf{E}_{L-1} = [\mathbf{E}_{L-1}^{class}, \mathbf{E}_{L-1}^{\mathbf{ids}(1)}, \mathbf{E}_{L-1}^{\mathbf{ids}(2)}, \cdots, \mathbf{E}_{L-1}^{\mathbf{ids}(k)}]. \quad (5)$$

By replacing the original input sequence of the last transformer layer with $\mathbf{E}_{L-1}$, the proposed SVF encourages the image encoder to highlight the subtle discrepancies between different subcategories while ignoring regions such as background or less meaningful foreground regions, thereby promoting fine-grained understanding.

## 3.3 Discriminative Model Training Strategy

With the collaborative efforts of the dual visual filtering mechanism, the proposed DVF demonstrates exceptional retrieval performance in FGIR tasks. However, it focuses on model design and does not effectively address the challenge of limited fine-grained image data, as noted in **G3**. Therefore, following **G3**, we delve into the formulation of discriminative model training strategy (DMT). Data augmentation is a training strategy that has proven effective in the field of classification [3, 7, 33] to alleviate the problem of limited training samples. Inspired by this, we incorporate data augmentation into the retrieval tasks as a component of DMT. It enhances data diversity throughout model training and diminishes the model's reliance on specific features from the training set. Specifically, we consider the following transformations:

- Color-jitter: Add perturbation to colors.
- Grayscale: Give more focus on shapes.
- Gaussian Blur: Slightly change the details in the image.

For each image, we consistently apply color-jitter and randomly apply Grayscale and Gaussian Blur. In Table 1, we present an ablation study of our various data augmentation components in the open-set setting.

Meanwhile, contrastive loss [8] is presented as a loss constraint into DMT, which improves the generalization and balance of the model in the training process. Formally, the contrastive loss with batch size $B$ is expressed as:

$$
\mathcal{L}_{con} = \frac{1}{B^2} \sum_{i=1}^{B} [ \sum_{j:y_i=y_j}^{B} (1 - \text{Sim}(\mathbf{E}_i^R, \mathbf{E}_j^R) +
$$
$$
\sum_{j:y_i \neq y_j}^{B} \max((\text{Sim}(\mathbf{E}_i^R, \mathbf{E}_j^R) - \beta), 0)], \tag{6}
$$

where $y_i$ is a subcategory label, $\text{Sim}(\mathbf{E}_i^R, \mathbf{E}_j^R)$ represents the dot product between $\mathbf{E}_i^R$ and $\mathbf{E}_j^R$, and the $\beta$ is a constant margin.

## 3.4 Overall Function

We optimize a training objective as below:

$$
\mathcal{L} = \mathcal{L}_{pnca} + \mathcal{L}_{con}, \tag{7}
$$

where $\mathcal{L}_{pnca}$ denotes the ProxyNCA loss [24]:

$$
\mathcal{L}_{pnca} = -\log \left( \frac{\exp(-d(\|\mathbf{E}_i^R\|, \|\mathbf{c}_i\|))}{\sum_{\mathbf{c} \in \mathbf{C}} \exp(-d(\|\mathbf{E}_i^R\|, \|\mathbf{c}\|))} \right), \tag{8}
$$

where $d(\|\mathbf{E}_i^R\|, \|\mathbf{c}_i\|)$ represents the distance between $\|\mathbf{E}_i^R\|$ and $\|\mathbf{c}_i\|$, $\mathbf{c}_i$ denotes the class proxy corresponding to retrieval embedding $\mathbf{E}_i^R$, $\mathbf{C}$ is the class proxy set, and $\| \cdot \|$ denotes the $L^2$-Norm.

## 4 EXPERIMENTS

## 4.1 Experiment Setup

*4.1.1 Datasets.* **CUB-200-2011** [2] comprises $11,788$ bird images from 200 bird species. In the closed-set setting, the dataset is divided into training and testing subsets comprising 5,994 and 5,794 images, respectively, out of a total of 11,788 images. For the open-set setting, we employ the first 100 subcategories (5,864 images) for training, and the remaining subcategories (5,924 images) are used for testing.

**Stanford Cars** [14] consists of 16,185 images depicting 196 car variants. Similarly, these images were split into 8,144 training images and 8,041 test images in the closed-set setting. For the open-set setting, we utilize the first 98 subcategories (comprising 8,054 images) for training and the remaining 98 subcategories (comprising 8,154 images) for testing.

**NABirds** [10] contains 48,562 images showcasing North American birds across 555 subcategories. For the closed-set setting, the training set contains 23,929 images, while the remaining 24,633 images are used for testing. For the open-set setting, we set up a more challenging training/test set split, with 20,984 images from the first 255 subcategories used for training and 27,578 images from the remaining subcategories used for testing.

*4.1.2 Evaluation protocols.* To evaluate retrieval performance, we adopt Recall@K with cosine distance in previous work [28], which calculates the recall scores of all query images in the test set. For each query image, the top **K** similar images are returned. A recall score of 1 is assigned if at least one positive image among the top **K** images; otherwise, it is 0.

*4.1.3 Implementation details.* In our experiments, we employ the ViT-B-16 [5] pre-trained on ImageNet21K [25] as our image encoder. All input images are resized to $256 \times 256$, and crop them into $224 \times 224$. In the training stage, we utilize the Adam optimizer and employ cosine annealing as the optimization scheduler. The initialize the learning rate as 3e-2 for all datasets. The number of training epochs is set to 10 for both CUB-200-2011 and Stanford Cars, while the NABirds are trained for 5 epochs, and the batch size is set to 32. The results in Table 4 are provided by the original paper results, while the results in Tables 2 and 3 were reproduced using the source code of the original paper.

## 4.2 Comparison with State-of-the-Art Methods

*4.2.1 Closed-set Setting.* We compare our proposed DVF with previous competitive methods in a closed-set setting. The comparison results for the CUB-200-2011 and Stanford Cars datasets are presented in Table 2, and the results for the NABirds dataset can be found in Table 3. From these tables, it can be observed that our proposed method outperforms other state-of-the-art methods on CUB-200-2011 and NABirds, and achieves competitive performance on Stanford Cars.

Specifically, in comparison with Hyp-ViT [6], the current state-of-the-art on CUB-200-2011, our DVF demonstrates a 3.4% improvement in Recall@1 and a 4.3% improvement over our base framework ViT [5]. The experimental results on the Stanford Cars indicate that our method outperforms the most of existing methods but falls slightly behind HIST [18]. We argue the possible reason may be attributed to the relatively regular and simpler shape of the cars. In the larger and more challenging NABirds dataset, we observe a significant superiority of the ViT structure over the CNN structure. Our DVF outperforms the best CNN method by 9.0% on the Recall@1 and improves the performance of the previous leading method ViT [5] by 8.1%.

*4.2.2 Open-set Setting.* The open-set setting poses greater challenges compared to the closed-set setting due to the unknown subcategories in the test set. The experimental results for the CUB-200-2011 and Stanford Cars datasets are shown in Table 4, while the results for the NABirds dataset are provided in Table 3. The results reveal a consistent trend in both open and closed-set settings: our method outperforms other state-of-the-art approaches on CUB-200-2011 and NABirds, while delivering competitive results on the Stanford Cars dataset.

To be specific, in comparison with Hyp-ViT [6] on CUB-200-2011, our DVF exhibits a 4.8% improvement in Recall@1 and a 6.2% enhancement over our base framework ViT [5]. Experimental results on the Stanford Cars dataset show that our method outperforms most existing methods but lags behind FRPT [35] by a slight margin. The results of experiments on NABirds are shown in Table 3. Even in more challenging settings, DVF consistently demonstrates excellent performance, whereas other methods, particularly those based on CNNs, experience a significant drop in performance. In numerical terms, our DVF surpasses the top-performing CNN method by 24.8% in Recall@1 and enhances the performance of the previously leading method Hyp-ViT [6] by 9.0%. The significant performance improvement can be attributed to the combined effects of the ViT structure and our proposed guidelines in Section 1.

**Table 2: Comparison with state-of-the-art methods in the closed-set setting on CUB-200-2011 and Stanford Cars 196. The best result is shown in bold, and the second-best result is underlined.**

| Method | Backbone | CUB-200-2011 | | | | Stanford Cars 196 | | | |
|---|---|---|---|---|---|---|---|---|---|
| | | Recall@1 | Recall@2 | Recall@4 | Recall@8 | Recall@1 | Recall@2 | Recall@4 | Recall@8 |
| PNCA [24] | CNN | 67.6 | 76.7 | 83.9 | 89.1 | 75.4 | 84.4 | 89.9 | 93.5 |
| Proxy-Anchor [12] | CNN | 80.4 | 85.7 | 89.3 | 92.3 | 77.2 | 83.0 | 87.2 | 90.2 |
| HIST [18] | CNN | 75.6 | 83.0 | 88.3 | 91.9 | **89.2** | **93.4** | 95.9 | 97.6 |
| PNCA++ [31] | CNN | 80.4 | 85.7 | 89.3 | 92.3 | 86.4 | 92.3 | 96.0 | 97.8 |
| ViT [5] | ViT | 83.3 | 88.6 | 92.6 | 95.1 | 83.1 | 89.8 | 93.7 | 96.4 |
| Hyp-ViT [6] | ViT | 84.2 | 91.0 | 94.3 | 96.0 | 76.7 | 85.2 | 90.8 | 94.7 |
| **DVF (Ours)** | ViT | **87.6** | **92.6** | **95.1** | **96.8** | 88.2 | 93.1 | **96.3** | **98.1** |

**Table 3: Comparison with state-of-the-art methods in the closed-set and open-set settings on NABirds.**

| Method | Backbone | Closed-set Setting | | | | Open-set Setting | | | |
|---|---|---|---|---|---|---|---|---|---|
| | | Recall@1 | Recall@2 | Recall@4 | Recall@8 | Recall@1 | Recall@2 | Recall@4 | Recall@8 |
| PNCA [24] | CNN | 54.4 | 66.3 | 75.9 | 84.2 | 45.2 | 56.5 | 66.7 | 76.0 |
| Proxy-Anchor [12] | CNN | 77.5 | 83.2 | 87.0 | 90.0 | 54.3 | 64.9 | 74.5 | 82.2 |
| HIST [18] | CNN | 71.8 | 78.4 | 83.4 | 87.5 | 51.8 | 62.8 | 72.5 | 81.0 |
| PNCA++ [31] | CNN | 79.9 | 87.0 | 92.0 | 95.2 | 63.4 | 74.0 | 82.2 | 88.4 |
| ViT [5] | ViT | 80.8 | 87.1 | 91.1 | 94.1 | 78.6 | 85.2 | 89.3 | 92.4 |
| Hyp-ViT [6] | ViT | 80.2 | 87.6 | 92.5 | 95.6 | 79.2 | 86.7 | 92.1 | 95.3 |
| **DVF (Ours)** | ViT | **88.9** | **93.5** | **96.3** | **97.9** | **88.2** | **93.0** | **96.1** | **97.8** |

**Table 4: Comparison with state-of-the-art methods in the open-set setting on CUB-200-2011 and Stanford Cars 196.**

| Method | Backbone | CUB-200-2011 | | | | Stanford Cars 196 | | | |
|---|---|---|---|---|---|---|---|---|---|
| | | Recall@1 | Recall@2 | Recall@4 | Recall@8 | Recall@1 | Recall@2 | Recall@4 | Recall@8 |
| PNCA [24] | CNN | 49.2 | 61.9 | 67.9 | 72.4 | 73.2 | 82.4 | 86.4 | 88.7 |
| DGCRL [45] | CNN | 67.9 | 79.1 | 86.2 | 91.8 | 75.9 | 83.9 | 89.7 | 94.0 |
| DAS [20] | CNN | 69.2 | 79.3 | 87.1 | 92.6 | 87.8 | 93.2 | 96.0 | 97.9 |
| IBC [26] | CNN | 70.3 | 80.3 | 87.6 | 92.7 | 88.1 | 93.3 | 96.2 | 98.2 |
| Proxy-Anchor [12] | CNN | 71.1 | 80.4 | 87.4 | 92.5 | 88.3 | 93.1 | 95.7 | 97.0 |
| HIST [18] | CNN | 71.4 | 81.1 | 88.1 | - | 89.6 | 93.9 | 96.4 | - |
| PNCA++ [31] | CNN | 70.1 | 80.8 | 88.7 | 93.6 | 90.1 | 94.5 | 97.0 | 98.4 |
| FRPT [35] | CNN | 74.3 | 83.7 | 89.8 | 94.3 | **91.1** | **95.1** | **97.3** | 98.6 |
| ViT [5] | ViT | 82.6 | 88.7 | 92.2 | 94.3 | 86.6 | 92.5 | 96.0 | 97.9 |
| DFML [34] | ViT | 79.1 | 86.8 | - | - | 89.5 | 93.9 | - | - |
| Hyp-ViT [6] | ViT | 84.0 | 90.2 | 94.2 | 96.4 | 86.0 | 91.9 | 95.2 | 97.2 |
| **DVF (Ours)** | ViT | **88.8** | **93.1** | **95.3** | **96.5** | 90.2 | 94.6 | **97.3** | **98.9** |

## 4.3 Ablation Studies and Analysis

*4.3.1 Efficacy of various components.* The proposed DVF comprises three essential components: OVF and SVF from the dual visual filtering mechanism, and Discriminative model Training strategy (DMT). We conducted ablation experiments on these components, and the results are reported in Table 5, with the Baseline representing the pure ViT. The introduction of SVF and OVF improved the Recall@1 performance on the CUB-200-2011 by 1.7% and 3.6% respectively. Besides, the combined use of OVF and SVF results in further performance improvement. This observation indicates that both OVF

and SVF can help DVF generate discriminative features, which are complementary, thereby improving the performance. Moreover, incorporating DMT into the model variation improved its performance on the CUB-200-2011.

*4.3.2 Number of selected embedding tokens in SVF.* The performance comparison of different values of $k$, representing the number of selected embedding tokens in SVF, is presented in Fig. 4. The performance gradually increases with the number of embedding tokens but decreases when the number exceeds 12. The decline in performance may be attributed to the fact that as the number

**Table 5: The Recall@K results (%) of component ablation study on CUB-200-2011.**

| Setting | R@1 | R@2 | R@4 | R@8 |
|---|---|---|---|---|
| Baseline | 82.6 | 88.7 | 92.2 | 94.3 |
| Baseline + SVF | 84.3 | 90.5 | 93.7 | 95.8 |
| Baseline + OVF | 86.2 | 91.2 | 93.8 | 95.5 |
| Baseline + OVF + SVF | 88.0 | 92.4 | 95.0 | 96.3 |
| Baseline + OVF + SVF + DMT | **88.8** | **93.1** | **95.3** | **96.5** |

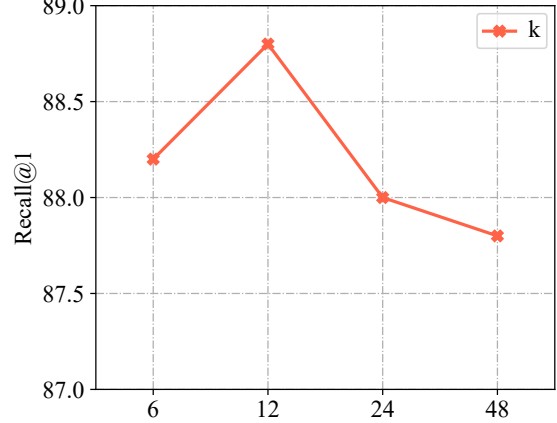

**Figure 4: Analyses of hyper-parameter $k$ on CUB-200-2011.**

**Table 6: The Recall@K results (%) for various methods with/without OVF on CUB-200-2011.**

| Method | R@1 | R@2 | R@4 | R@8 |
|---|---|---|---|---|
| PNCA++ [31] | 70.1 | 80.8 | 88.7 | 93.6 |
| PNCA++ w/ OVF | 72.4↑2.3 | 82.4↑1.6 | 89.4↑0.7 | 93.8↑0.2 |
| ViT [5] | 82.6 | 88.7 | 92.2 | 94.3 |
| ViT w/ OVF | 86.2↑3.6 | 91.2↑2.5 | 93.8↑1.6 | 95.5↑1.2 |
| DVF (Ours) w/o OVF | 85.7 | 91.0 | 94.2 | 96.0 |
| DVF (Ours) | 88.8↑3.1 | 93.1↑2.1 | 95.3↑1.1 | 96.5↑0.5 |

$k$ rises, the semantic visual filtering may emphasize the locations of objects or parts rather than focusing on subcategory-specific discrepancies, making it ineffective for FGIR. Consequently, we have chosen to set $k$ to 12 for all datasets.

*4.3.3 Generalizability of OVF.* The proposed OVF is training-free and model-agnostic. Therefore, we conducted comparative experiments to explore the generalizability of OVF. The results in Table 6 show that the performance of both the CNN-based method and the ViT-based method is significantly improved after integrating OVF. This demonstrates the generality of OVF, as it can aid various methods to emphasize the object, allowing them to concentrate more on subcategory-specific discrepancies and consequently improving performance on FGIR.

**Table 7: Comparison of Recall@K results (%) on CUB-200-2011 with/without token importance generator $\Omega$.**

| Setting | Recall@1 | Recall@2 | Recall@4 | Recall@8 |
|---|---|---|---|---|
| w/o $\Omega$ | 87.5 | 92.3 | 95.1 | 96.3 |
| w/ $\Omega$ | **88.8** | **93.1** | **95.3** | **96.5** |

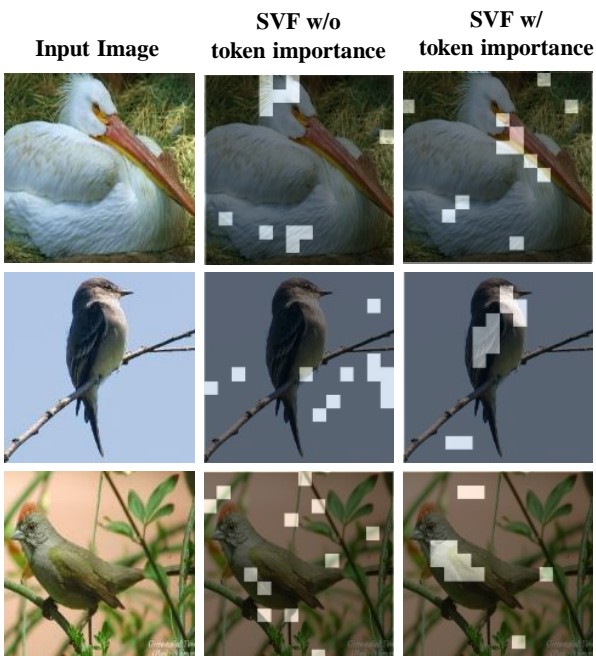

**Figure 5: Visualization of SVF with/without token importance on CUB-200-2011. The token importance empowers DVF to concentrate on discriminative regions, including the beak, tail, and areas with color mutations.**

*4.3.4 Importance of the Token Importance Generator.* Here, we investigate the influence of the token importance generator on retrieval performance. The retrieval results, presented in Table 7, demonstrate that the performance declines in the absence of the token importance generator. The possible reason for the performance degradation is that, without the token importance generator, SVF can generate semantic scores solely based on class token; however, its ability to capture discriminative regions is limited.

To more intuitively reflect the significance of the token importance generator, we conducted a visual experiment to demonstrate its effectiveness, as depicted in Fig. 5. The first column represents the input image, while the second and third columns illustrate the selected tokens without/with token importance, respectively. Visualization results show that the token importance generator enables DVF to focus more on discriminative areas such as the beak, tail, and color mutation areas.

*4.3.5 Influence of DMT.* To demonstrate the effectiveness of the components in the proposed DMT. We conducted ablation experiments on DMT components, and the results can be found in Table 8.

**Query image**

**Top-10 retrieved images**

Figure 6: Examples of top-10 retrieved images on CUB200-2011 by DVF. The images with green boxes are the correct ones, and those with red boxes are the wrong ones.

Table 8: Ablation of DMT components on CUB-200-2011.

| Setting | Recall@1 | Recall@2 |
|---|---|---|
| DVF w/o DMT | 88.0 | 92.4 |
| DVF w/o data augmentation | 88.3 | 92.8 |
| DVF w/o contrastive loss | 88.4 | 92.9 |
| DVF | **88.8** | **93.1** |

It is observed that both data augmentation and contrastive loss contribute to performance improvement, proving the effectiveness of DMT components in training discriminative models. Furthermore, the DMT achieves the best results, demonstrating that data augmentation and contrastive loss are complementary to each other.

*4.3.6 Visualization.* Upon visualizing the top 10 results obtained from CUB-200-2011 in Fig. 6, we observe that DVF excels in retrieving images belonging to the same subcategory across various subcategories, even amidst diverse variations and backgrounds. There is also a failure case that requires careful observation of the subtle differences between the query image and the returned image.

Additionally, we conduct visualization experiments to illustrate the effectiveness of DVF, as shown in Fig. 7. The original input images are presented in the first column, followed by the second column illustrating the class activation maps computed using Grad-CAM for the baseline model, which takes the original input images as input. The third column describes the input image with OVF, while the fourth column shows the class activation map of DVF taking it as input. The visualization results demonstrate that DVF effectively avoids focusing on background regions and enhances the baseline model's ability to focus on a more comprehensive region, especially in capturing more detailed discrepancies specific to subcategories, such as head, wings, and tail.

## 5 CONCLUSION

In this paper, we propose a set of guidelines for designing fine-grained image retrieval (FGIR) models by analyzing the unique

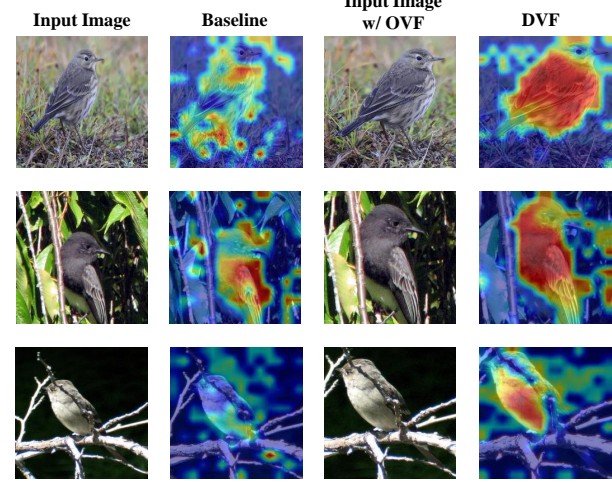

Figure 7: Class activation visualizations on CUB-200-2011. For each sample, from left to right, we show the input image, class activation map of the baseline model, input image with OVF, and the class activation map of the proposed DVF. We can observe that DVF captures more comprehensive regions.

characteristics of FGIR tasks and the shortcomings of previous methods. Following these guidelines, we propose an effective FGIR model and a discriminative training strategy. Specifically, we propose a visual transformer with an object-oriented visual filtering module, and a semantic-oriented visual filtering module to generate discriminative representation. Furthermore, we introduce a discriminant model training strategy that combines data augmentation with contrastive loss to enhance the discriminative and generalization capabilities of the model. The effectiveness of the proposed guidelines is verified through extensive ablation studies. The experiment results demonstrate that our model, following these guidelines, achieves state-of-the-art performance on three challenging fine-grained image retrieval benchmarks.

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
