# OpenReview forum: "DVF: Advancing Robust and Accurate Fine-Grained Image Retrieval with Retrieval Guidelines"
_acmmm.org/ACMMM/2024/Conference — MM2024 Poster_

### Official Review · Reviewer_YFaE · 2024-05-11

**Rating:** 5
**Confidence:** 4

**Summary:**

The paper presented object recognition guidelines for fine-grained image retrieval (FGIR) tasks. In addition, following the proposed guidelines, the author introduced a novel Dual Visual Filtering mechanism (DVF) and achieved notable performance gain compared with some benchmarks. The proposed framework is designed for transformer architecture but the object location and augmentation concept can be utilised for other architecture or tasks.

**Strengths:**

The authors presented and followed a set of guidelines for designing high-performance fine-grained image retrieval models, which makes their method clear to follow.

The proposed object-orient visual filtering module is model-agnostic, which can be easily used on other classification tasks to augment input images.

The token importance generator in the semantic-oriented visual filtering module can locate discriminative regions.

**Limitations:**

1. The CCS concept after the abstract section is not generated correctly.

2. The benchmark comparison methods with ViT backbone in Tables 2-4 are not state-of-the-art and not enough. Should include more state-of-the-art methods based on ViT to allow better understanding in the related field.

3. The object-orient visual filtering module adopts an object detection model (Grounding-DINO) to locate the target object, which consists of text encoding and image encoding. Even though it’s training-free, the detection and post-processing operations may still cause significant processing time and computational power. The author should provide training details for this consideration.

4. The post-processing operation requires a certain amount of image adjustment to maintain the object intact and prevent deformation, which makes it hard to find the optimal parameters since the objects in the image can have different ratios and sizes.

5. The author stated that the semantic score extracted by the transformer architecture can raise the issue of inaccurate subcategory prediction (line 398). However, there’s no evidence to prove it.

6. The ablation study of Table 1 should be placed after the main experiments (Tables 2-4).

**Suitability:**

2

---

### Official Review · Reviewer_F2j7 · 2024-05-20

**Rating:** 4
**Confidence:** 3

**Summary:**

This paper presents a meticulous analysis leading to the proposal of practical guidelines to identify subcategory-specific discrepancies and generate discriminative features to design effective FGIR models.

**Strengths:**

This paper formulates a set of practical guidelines for designing high-performance fine-grained image retrieval models and develops a straightforward yet potent retrieval model, called DVF, based on the proposed guidelines. This paper has a clear motivation and is well written.

**Limitations:**

(1) There is no mention of Figure 1 in the introduction or even in the entire paper, so what is the role of Figure 1?
(2) The related work is too brief for me to get a clear indication of the relevant developments in the techniques utilized by the authors.
There are some issues in the experimental parts that confuse me.
(3) The backbone network utilized by the authors, ViT-B-16, is pre-trained on ImageNet21K, while the backbone network utilized by the state-of-the-art FRPT, ResNet50, is pre-trained on ImageNet1K, so is this a fair comparison? If ViT-B-16 is pre-trained on ImageNet1K, whether the performance still superior?
(4) On the Stanford Cars dataset, DVF performance falls slightly behind HIST. The authors argue that the possible reason may be attributed to the relative regularity and simple shape of the cars.  However, the essence of fine-grained image retrieval is to discriminate detailed features, so how does the relative regularity and simple shape of the objects affect the performance of the retrieval?
(5) In the open-set setting on the Stanford Cars dataset, DVF performance falls slightly behind FRPT, while the remaining two datasets are birds. Therefore, whether the model has strong generalization ability?
(6) For the dataset utilized in the paper, each image contains one instance; If each image contains multiple instances (e.g., VegFru dataset and Food101 dataset), whether DVF can deal with it?

**Suitability:**

3

---

### Official Review · Reviewer_i9wZ · 2024-05-23

**Rating:** 5
**Confidence:** 2

**Summary:**

The paper presents an approach to enhance fine-grained image retrieval (FGIR) through a novel method called Dual Visual Filtering (DVF). The paper establishes practical guidelines to differentiate subcategory-specific discrepancies and proposes a novel visual transformer mechanism that leverages these guidelines to improve feature discriminability and retrieval accuracy across closed-set and open-set scenarios.

**Strengths:**

1.Theoretical and Practical Contributions: The paper introduces practical guidelines that contribute both theoretically and technically to the FGIR field. These guidelines—focusing on emphasizing the object, highlighting subcategory-specific discrepancies, and employing effective training strategies—provide a structured framework that advances current methodologies.
2. Novelty of the Dual Visual Filtering Mechanism: The DVF model, which incorporates an object-oriented and a semantic-oriented visual filtering module, is a significant innovation. This dual approach allows for enhanced discrimination of fine-grained features, which is crucial for tasks requiring high levels of detail.
3. Empirical Validation: The empirical studies are comprehensive, involving extensive ablation studies and comparisons with state-of-the-art methods. The use of widely-recognized datasets like CUB-200-2011, Stanford Cars, and NABirds, and the detailed performance metrics provided, substantiate the effectiveness of the proposed approach.
4. Clarity and Quality of Presentation: The paper is well-organized and clearly written, with detailed methodological explanations and a logical flow that aids in understanding the complex processes involved in the proposed FGIR approach.

**Limitations:**

1. Lack of Diverse Testing Scenarios: While the paper provides extensive results, the testing primarily focuses on standard datasets. Additional testing in more varied real-world scenarios or across more diverse datasets could further validate the robustness of the proposed model.
2. Computational Efficiency: The paper does not fully address the computational implications of the Dual Visual Filtering mechanism. Given the complexity of the approach, it could be resource-intensive, which might limit its applicability in resource-constrained environments.

**Suitability:**

2

---

### Official Review · Reviewer_LpvN · 2024-05-24

**Rating:** 2
**Confidence:** 4

**Summary:**

The paper provides guidelines for designing effective Fine-Grained Image Retrieval (FGIR) models. It introduces a Dual Visual Filtering (DVF) mechanism, composed of object-oriented and semantic-oriented modules, to capture subcategory-specific discrepancies. The adjustment of training strategy enhances the DVF's discriminative and generalizability. Through extensive analysis, DVF showed notable performance on three fine-grained datasets.

**Strengths:**

1. The method proposed in this paper was verified on fine-grained image retrieval datasets, and ablation experiments were carried out, proving the method is effective.
2. The writing of this article is concise and easy to understand.

**Limitations:**

1. The method lacks innovation. OVF (Object Visual Filtering): The idea of using a detection model to crop image objects is fairly common. SVF (Semantic Visual Filtering): Re-weighting of image patch features is also ordinary. DMT (Discriminative Model Training): This is essentially data augmentation, and compared to common data augmentation, there's no new idea.

2. The applicability is very limited. Although it can better differentiate fine-grained object categories, it only works on well-curated datasets and is not suitable for slightly complicated real-world scenarios, such as images with multiple target objects. The open-set setting mentioned in the paper cannot demonstrate the method's generalizability, because even when applied to an open-set, it's limited to objects of the same kind, such as "bird" or "car".

3. Advices (not limitations, might worth trying): The authors could start with image-text pair data, fully leveraging the capabilities of the open-set detection model, and crop multiple objects using words from the text. Then, they could perform re-weighting not only on patches of each object but also among multiple objects. This approach would both accommodate fine-grained categories and adapt to complex scenarios, as well as allow for scaling up.

**Suitability:**

3

---

### Meta-Review · Area_Chair_6MYJ · 2024-07-01

**Recommendation:** Accept (Poster)
**Confidence:** 5

**Metareview:**

This paper has received two WAs, one BA, and one WR as initial scores.

Pros:
All reviewers agree that the paper is well written and easy to understand.
The empirical studies are comprehensive, involving extensive ablation studies and comparisons with state-of-the-art methods.
Reviewers i9wZ and F2j7 recognize the novelty of DVF.

Cons:
Reviewer LpvN raises concerns about the use of a detection model to crop image objects, which is a fairly common practice.
Reviewers i9wZ F2j7, and YFaE have concerns about the experiments.

The authors provide a rebuttal which addresses the reviewers' concerns. The AC agrees with the majority of reviewers that the paper should be accepted.